# Synthesising turn-taking cues using natural conversational data

*Johannah O'Mahony[1], Catherine Lai[1], Simon King[1]*

[1]Centre for Speech Technology Research, United Kingdom

`johannah.o'mahony@ed.ac.uk`

## Abstract

As speech synthesis quality reaches high levels of naturalness for isolated utterances, more work is focusing on the synthesis of context-dependent conversational speech. The role of context in conversation is still poorly understood and many contextual factors can affect an utterances's prosodic realisation. Most studies incorporating context use rich acoustic or textual embeddings of the previous context, then demonstrate improvements in overall naturalness. Such studies are not informative about what the context embedding represents, or how it affects an utterance's realisation. So instead, we narrow the focus to a single, explicit contextual factor. In the current work, this is turn-taking. We condition a speech synthesis model on whether an utterance is turn-final. Objective measures and targeted subjective evaluation are used to demonstrate that the model can synthesise turn-taking cues which are perceived by listeners, with results being speaker-dependent.

**Index Terms**: dialogue, context-aware TTS, turn-taking

## 1. Introduction

Text-to-Speech (TTS) naturalness is often very high for isolated read speech utterances. But, speech is rarely produced in isolation: it is spoken in *context*. This is particularly true for conversation. Recent work has focused on synthesis of contextualised conversational speech by incorporating information about the previous acoustic and/or textual context [1, 2, 3, 4, 5]. Context is a broad term and has different effects on an utterance e.g., pragmatic context [6], entrainment with a speaking partner or utterance position in a turn [7]. The effect of context on speech, especially its prosodic realisation, is poorly understood and can be data-dependent [8]. Prosodic cues also show inter-speaker variability [9]. This makes it difficult to predict what effect context should have on an utterance, and to measure progress if using paradigms such as Mean Opinion Score naturalness.

All of the above methods involve conditioning synthesis on a pre-trained embedding such as BERT for text, or an acoustic embedding of previous acoustic content. It is unclear what these representations capture and, importantly, whether they capture *individual* effects reported in the literature. To better understand what can be learned by representing context and how different aspects of context affect speakers' realisations, it is important to test *individual* contextual factors in isolation and to create suitable evaluation paradigms for them. This will not only provide better evidence for the effect of context in natural speech, but will allow us to make hypotheses about expected speech synthesis model behaviour.

We therefore narrow our focus to a single aspect of context, to observe its effect on prosodic realisation. While most work has focused on the *prior* context, here we condition the model on whether a speaker continues talking or gives up their turn. This is known as turn-taking and is a key aspect of conversation which affects how speakers realise an utterance [7]. In contrast to the work cited above, we use natural conversational speech, not (semi-)scripted dialogue. We answer three questions:

1) Does conditioning a TTS model on turn-taking enable the model to learn prosodic turn-taking cues from natural conversational speech that are perceptible to listeners? 2) Does conditioning a TTS model on turn-taking lead to increased turn-finality judgements over a baseline? 3) What prosodic cues distinguish natural turn-medial vs turn-final utterances and are they also present in synthetic speech?

## 2. Previous Work

Turn-taking is an important aspect of interaction [10] that helps speakers signal to their partner whether they are giving up their speaking turn or have more to say. The signal can be any combination of pragmatic and/or syntactic and/or prosodic cues, all optional, but which have an additive effect [10, 11].

Most work on context uses datasets comprising scripted or semi-scripted conversation. This might not contain the true-to-life interactional turn-taking features found in natural conversational speech. Such datasets are often constructed for back-and-forth agent-user interaction, and so include an unnaturally-high proportion of turn-final utterances. In corpus research on spontaneous speech, the usual unit of analysis is the inter-pausal unit (IPU). A conversational turn can comprise many IPUs so there is a class imbalance between turn-final and turn-medial IPUs. If training a TTS model on natural conversational speech, we are likely to have far more turn-medial IPUs in the training data, which might potentially lead to more turn-medial prosody. In this study, we use natural conversational speech to train our TTS models. We are not the first to do this, but prior work does not consider context, e.g., [12, 13, 14, 15].

Specific prosodic and acoustic turn-taking phenomena have been found in English corpus research [16] and prosodic cues have been found to be important for turn-end estimation [17]. Various cues have been found to be different between turn-final and turn-medial transitions in the literature including creaky voice [18], shorter turn-final IPUs, final-word lengthening, and increase in speech rate [16, 7], as well as differences in F0 [7].

## 3. Method

### 3.1. Data

#### 3.1.1. Conversational Data

We use the CANDOR Corpus [19], a corpus comprised of 1656 open-domain online conversations between two speakers,

recorded in separate channels, and transcribed automatically. To split into utterances, we implemented communicative state classification [20] in which we split each channel into IPUs at every silence longer than 200 ms, dropping IPUs comprising only backchannels to reduce the chance of overlapping speech (out of scope here). We selected target IPUs if no overlap occurred on either side leaving 262 446 [left-context]-[target]-[right-context] triplets, labelled according to who spoke in the right-context IPU, i.e., *same* vs *different* speaker to the target.

For initial data selection we chose target IPUs with duration 1 s to 11 s surrounded by no more than 2 s of silence on either side. All targets with text containing symbols, numbers or acronyms were removed. We removed targets containing a question-mark (a proxy for questions; our focus is declarative prosody). We then stripped all automatically-inserted punctuation. We calculated speech rate using canonical syllables per second and only retained target IPUs with speech rate 2-6 syllables per second. We finally removed triplets where the left-context or target IPU had fewer than 3 words.

We then made two subsets of data. The first (modelling) consisted of all speakers with more than 10 min of speech, to be used for modelling. The second (heldout) contained speakers with less than 10 min of speech, to be used as a source of text for our evaluation, and never used in training. After initial filtering we had 51 093 targets (turn-medial 40 739; turn-final 10 354) totalling 56.72 h. We aligned the modelling data using the Montreal Forced Aligner (MFA) [21] and removed all targets where alignment failed and all speakers where more than 5% of their utterances failed.

### 3.1.2. Read Speech Data

Initial tests showed that training models on only the above data was not feasible, due to data quantity and variable recording conditions. Therefore the LJSpeech corpus was used in addition, which we chunked at punctuation (which was then stripped) to shorten the utterances and make them more comparable to IPUs in the conversational data. We removed utterances with fewer than 3 words, leaving 16 000 utterances for training and 100 for development.

### 3.1.3. Final Data Selection

Natural conversations exhibit class imbalance – there are far more turn-medial IPUs – so further selection was required to improve class balance. We first took all data from the 41 speakers with 15 min to 20 min of speech. To this we added all data from speakers with a high number of turn-final IPUs, plus all turn-final IPUs from the remaining speakers.

We then calculated the number of turn-final IPUs per speaker and, by random selection, ensured the data contained the same number of turn-medial IPUs for that speaker. The resulting dataset contains all available turn-final IPUs, but still has some class imbalance: Table 1. We then partitioned into 24 000 utterances (23 914 CANDOR + 86 LJSpeech) for training, 100 for development, and the remaining 242 for testing. There are 212 unique speakers (including LJ).

### 3.1.4. Training Data Acoustic Analysis

F0 and intensity contours were extracted using Praat at 10 ms intervals. F0 parameter settings were automatically determined [22], with per speaker floor and ceiling based on global raw F0 values. F0 and intensity were normalised to make values comparable across speakers and samples. Intensity was normalised

by subtracting speaker mean per IPU. F0 was converted to semitones relative to speaker global mean in Hz. Values more than 2.5 standard deviations from the utterance mean were removed as outliers. We also checked for octave jumps ending outside of the 5th and 95th quantiles per IPU and removed values outside of those.

We characterised F0 and intensity contours using using Legendre Polynomial (LP) decomposition. LP coefficients have been shown to provide interpretable characterisations of F0 contours [23]: e.g., the first 3 LP coefficients can be called height, slope, and convexity of the contour. Coefficients were determined using least squares fit of an order 5 Legendre series over a specified interval, time normalized to span [-1,1]. To analyse potential differences in turn-medial and turn-final prosody, we calculate LP coefficients for F0 and intensity over the last 500 ms of the IPUs in the training data, inspecting only the first 3 coefficients. We also calculate the speech rate (syllables/second) over the whole IPU based on phone alignments. See Table 1.

Table 1: *Descriptive statistics and median acoustic feature values for final conversational corpus.*

|  | Turn-Medial | Turn-Final |
|---|---|---|
| Total Turns | 15788 | 8468 |
| Duration (h) | 17.99 | 8.74 |
| Mean Tokens | 13.00 | 12.03 |
| Female Utterances | 5740 | 3002 |
| Male Utterances | 10 048 | 5466 |
| *Acoustic Features* | | |
| F0 height (LP coeff 1) | -1.05 | -1.12 |
| F0 slope (LP coeff 2) | -0.40 | -0.15 |
| F0 convexity (LP coeff 3) | 0.13 | 0.18 |
| Intensity height (LP coeff 1) | -0.59 | -0.76 |
| Intensity slope (LP coeff 2) | -2.56 | -3.50 |
| Intensity convexity (LP coeff 3) | -2.44 | -2.99 |
| Speech rate (syll/s) | 4.00 | 4.10 |

We found significant differences in the first 3 LP coefficients for F0 and intensity, as well as speaking rate (Wilcoxon ranked sum test, $p < 0.05$). Overall, turn-final IPU ends are characterised by a lower, flatter F0 contour and lower overall intensity. We observe a slightly faster speaking rate in the turn-final condition, but differences between conditions are small.

### 3.2. Models

We trained a **TURN** and a **BASELINE** FastPitch [24] model with identical architectures. We adapted the FastPitch model to use phone durations from MFA as input to the duration predictor during training and used the automatically aligned phoneme sequence as our input to the model (including silence tokens). We applied global mean/variance per-speaker F0 normalisation. Turn-conditioning is incorporated using an embedding table, whose output is summed to the encoder output and speaker embedding, before being passed forward to the variance adapters. The turn-condition input has three possible values: 0 for baseline; 1 for turn-medial; 2 for turn-final.

The only difference between **BASELINE** and **TURN** is the value of the turn-condition input. For **BASELINE**, it is fixed to 0 throughout all training and during inference. For **TURN**, it is set to 0 for read-speech data, to 1 for conversational IPUs labelled as turn-medial, and to 2 for conversational IPUs labelled as turn-final.

Pre-training was identical for both models, using only the read speech data with a batch size of 16 for a total of 200k steps. For both **BASELINE** and **TURN**, we completed training with a batch size of 32 for an additional 525k steps using the conversational data described above, plus 86 LJSpeech utterances. For both models the speaker embedding table contained 500 speaker codes and the turn-condition embedding table 3 codes. Each was trained on on a single NVIDIA GeForce GTX 1080 Ti GPU. After FastPitch inference, waveforms were generated using the HiFi-GAN universal vocoder [25] with a denoising factor of 0.01.

# 4. Subjective Evaluation

Most previous studies demonstrate some overall improvement in naturalness, using standard testing procedures like Mean Opinion Score. This does not provide any insight into what listeners were attending to. Instead, we use a method inspired by the work of [26].

## 4.1. Test Materials

### 4.1.1. Text Selection

Turn-taking cues are not only prosodic. In fact, the textual content is a strong cue, with prosody having an *additive* effect [10, 11]. Because of this, when evaluating prosodic turn-taking cues it is important to choose turn-neutral texts, where the likelihood of a turn-end cannot easily be judged using text alone [26]. We selected 600 random utterance texts from our held out data, 300 turn-final and 300 turn-medial, which were then filtered for personal identifying material, controversial topics and profanity, leaving 532 sentences. We divided these into 4 groups of 133. Each group was presented to 10 participants who were asked to rate each sentence (presented as text only) for turn-finality on a scale of 1-5 where a rating of 3 indicates maximum uncertainty that the speaker had finished talking. We used the instruction wording from [26]. We then took the median and mode rating of each text and chose utterances with only one mode, a mode of 3 and a median rating between 2.75-3.25.

### 4.1.2. Target Speaker Selection

As expected, given the large number of speakers with variable data quantity and recording conditions, the models could not synthesise all speakers with good quality. Expert listening to synthetic speech from the **BASELINE** model for the 53 speakers with more than 10 min of training speech was used to eliminate potential target speakers before the formal listening test, reducing the pool to 24 speakers. 20 participants then rated the naturalness of 5 synthetic utterances for each speaker on a scale of 1-5 (total utterances = 120), presented as speech only, in a randomised order. Using these ratings, we picked the best 5 target speakers (Table 2) for use in the following listening tests.

Table 2: *Target Speaker Corpus Training Information*

| Speaker | Total Utts | Turn-medial | Turn-final | Mean Naturalness Rating Pretest |
|---|---|---|---|---|
| 48 | 192 | 174 | 18 | 2.91 |
| 143 | 238 | 216 | 22 | 2.82 |
| 156 | 273 | 190 | 83 | 2.77 |
| 176 | 184 | 168 | 16 | 2.93 |
| 200 | 245 | 200 | 45 | 2.85 |

## 4.2. Participants

In all of the experiments, listeners were recruited using Prolific[1] and reported being English native speakers, residing in the US, with no hearing impairments. At the beginning of the survey, we asked participants if they were using headphones and at the end whether they could play all of the audio.

## 4.3. Statistical Analysis

For all AB experiments below we used binomial mixed-effects regression models with a logit-link function [27] due to lack of independence in listeners and stimuli. We included stimulus and listener as random effects. No predictors were included, making it a mixed-effects equivalent to an exact binomial test with the null hypothesis being that participant choice does not differ from chance. For each experiment, *choice* denotes the chosen audio for most final sounding and the model is specified as:

```
choice ~ 1 + (1|listener) + (1|stimulus)
```

## 4.4. Experiment 1 – Finality Judgements, Turn Model

To answer Question 1 (end of Section 1), we tested whether conditioning the **TURN** model on turn-medial vs. turn-final led to a perceptible difference for listeners. For each of the 5 target speakers, we synthesised two versions of the 50 turn-ambiguous[2] texts using the **TURN** model, one turn-medial, the other turn-final, then conducted a separate listening test for each. In each test, 20 listeners were presented with 50 pairs of synthetic utterances. The order was shuffled and the within-pair order randomised, per listener. We asked listeners *Which of the following sounds like the speaker is finished talking?*.

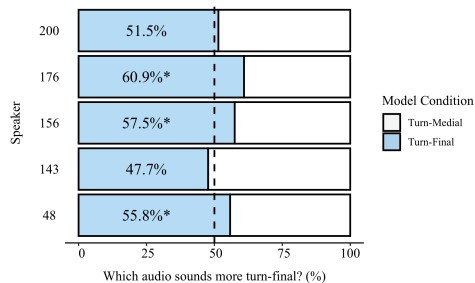

Figure 1: *Results for Experiment 1 comparing **TURN** generating turn-medial vs **TURN** generating turn-final*

20 participants took part in each listening test (total=100; of which we removed 3 for not wearing headphones and 3 for having issues playing audio). We analysed each listening test (i.e., each target speaker) independently and report the results in Table 3 and Figure 1. We found a significant majority choice for the turn-final synthesis for 3 speakers, with 2 speakers showing no significant difference in choice. This demonstrates that our model is able to learn patterns of turn-taking, but that the results are speaker-dependent.

[1]https://www.prolific.co
[2]Samples: https://johannahom.github.io/SSW-2023/

Table 3: *Results for Experiment 1 comparing **TURN** generating turn-medial vs **TURN** generating turn-final, per speaker*

| Speaker | $\beta$ | Probability Estimate | Confidence Interval | p-value |
|---|---|---|---|---|
| 48 | 0.25 | 0.56 | 0.51 - 0.61 | < 0.05 |
| 143 | -0.11 | 0.47 | 0.40 - 0.54 | > 0.05 |
| 156 | 0.33 | 0.58 | 0.52 - 0.64 | < 0.05 |
| 176 | 0.49 | 0.62 | 0.56 - 0.68 | < 0.05 |
| 200 | 0.06 | 0.52 | 0.46 - 0.57 | > 0.05 |

Table 4: *Results of Experiment 2 comparing **TURN** generating turn-final vs **BASELINE**, per speaker.*

| Speaker | $\beta$ | Probability Estimate | Confidence Interval | p-value |
|---|---|---|---|---|
| 48 | 0.34 | 0.58 | 0.50 - 0.66 | < 0.05 |
| 143 | 0.004 | 0.50 | 0.45 - 0.55 | > 0.05 |
| 156 | -0.03 | 0.49 | 0.42 - 0.56 | > 0.05 |
| 176 | 0.33 | 0.58 | 0.52 - 0.64 | < 0.05 |
| 200 | 0.31 | 0.58 | 0.53 - 0.63 | < 0.05 |

### 4.5. Experiment 2 – Finality Judgements, Turn Model vs Baseline

To answer Question 2 (end of Section 1), we need to test whether our **TURN** model in turn-final condition sounds more turn-final than **BASELINE**, which is trained on both turn-final and turn-medial IPUs. We synthesised the same 50 turn-ambiguous texts used in Experiment 1, but this time we compared the output of the **BASELINE** model with the **TURN** model operating with the turn-condition input set to turn-final. Again, we tested each speaker in a separate listening test, using the same design as Experiment 1.

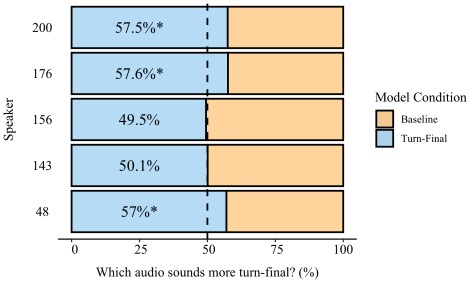

Figure 2: *Results of Experiment 2 comparing **TURN** generating turn-final vs **BASELINE***

20 participants took part in each listening test (total=100; of which we removed 5 for not wearing headphones and 4 for having audio issues). The results are summarised in Table 4. Here we can see that speaker 176 and 48 mirror Experiment 1: the turn-final condition sounds more turn-final than baseline. Speaker 143 remains on par with baseline. We see a change for speaker 200 who sounded more turn-final than baseline here, but who had no significant preference to turn-medial in Experiment 1. Speaker 156 on the other hand had a significant difference in Experiment 1, but now is no different to baseline; this might be due to this speaker having a high number of turn-final IPUs in the training data (Table 2), making **BASELINE** sound turn-final.

Overall results taken together indicate that **TURN** is able to produce turn-taking cues more effectively than **BASELINE**, for most but not all speakers.

## 5. Objective Evaluation

To investigate the cause of the speaker-dependent results from experiment 1 and 2, and to answer Question 3 (end of Section 1), we extracted the same acoustic information as for the training data. By analysing the acoustic output of each model condition along with the training data for each speaker, we hope to provide insight into which cues may be important for listen-

ers when determining turn-finality. We are also interested in whether cues found in the literature are also found in our turn-final synthesis. We hypothesise that speakers who show more differences in cues between conditions and should show large preferences in turn-finality ratings as cues should have an additive effect [11]. Measuring cues and comparing to the training data of each speaker can also provide insight into how much information is learned across speakers in the data and how much is constrained by speaker conditioning.

### 5.1. Target Speaker Acoustic Features Natural Speech turn-medial vs turn-final

Table 5 shows differences in acoustic features for turn-medial and final IPUs for our target speakers. Compared to the training data on aggregate, we observe speaker-specific cues for turn-taking: not all features show significant differences. Moreover, differences are not always in the same direction. For example, Figure 3 shows distribution of F0 height (1st LP coefficient) for target speakers. We see that Speaker 176 has a lower F0 for turn-final IPUs, while speaker 48 has higher F0. In fact, listening to samples suggests that speaker 48 has utterance final pitch rises as a default, i.e. uptalk. Speaker 48 also exhibited higher intensity (height) in the turn-final condition (though not significantly so), while the 4 other speakers showed lower intensity turn-finally. Mean speech rate was slower at the end of turn-medial IPUs for two speakers, however the differences were not significant for any of the speakers.

Table 5: *Significant differences ($p < 0.05$) between turn-medial and turn-final IPUs for speakers (Wilcoxon ranked sum test)*

| | 48 | 143 | 156 | 176 | 200 | |
|---|---|---|---|---|---|---|
| F0 height | ✓ | . | ✓ | ✓ | . | |
| F0 slope | . | . | . | . | . | |
| F0 convexity | . | ✓ | . | ✓ | . | |
| Intensity height | . | . | ✓ | . | . | . |
| Intensity slope | . | ✓ | ✓ | . | . | |
| Intensity convexity | . | . | . | ✓ | . | |
| Speech rate (syll/s) | . | . | . | . | . | |

### 5.2. Comparison of features turn-final and turn-medial

First for experiment one we compare the values of various prosodic markers found in Table 6 between the turn-medial and turn-final output per speaker. As we can see, across all speakers we find a significant difference between turn-medial and turn-final speech rate and final word duration. This mirrors results found in the study of [7]. In Figure 4 we can see the direction of this difference, with all speakers showing an increase in final word duration when synthesised as turn-final and speech rate being significantly faster than in turn-medial position. Interestingly, differences in these features do not directly lead to an increase in turn-final choices in experiment 1 as in the case

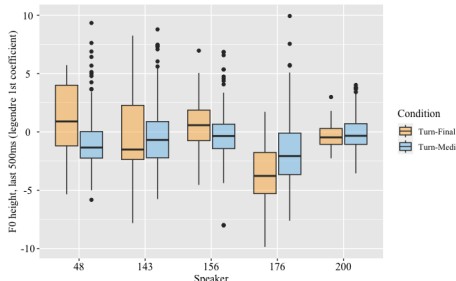

Figure 3: *F0 height per speaker, Last 500 ms*

of speaker 143 and 200. We found however, and if we look at the average pitch height of the final word (Figure 4) that speaker 143 had the phenomenon of uptalk, and this was particularly the case in turn-final position which may have led to more turn-medial judgements. Similarly, speaker 200 compared to the other speakers shows no significant difference between turn-final and turn-medial conditions for f0 height.

Table 6: *Significant differences ($p < 0.05$) between turn-medial and turn-final output for speakers (Wilcoxon signed-ranks test)*

| featnice | 48 | 143 | 156 | 176 | 200 |
|---|---|---|---|---|---|
| Global F0 height | ✓ | ✓ | . | ✓ | . |
| Global F0 slope | . | . | . | . | ✓ |
| Global F0 convexity | . | . | . | . | . |
| Final word F0 height | . | ✓ | ✓ | ✓ | . |
| Final word F0 slope | . | . | . | . | . |
| Final word F0 convexity | . | . | . | . | . |
| Intensity height | . | . | . | . | . |
| Intensity slope | . | . | . | . | . |
| Intensity convexity | . | ✓ | . | . | . |
| Final word log duration | ✓ | ✓ | ✓ | ✓ | ✓ |
| Speech rate (syll/s) | ✓ | ✓ | ✓ | ✓ | ✓ |

### 5.3. Comparison of features turn-final and baseline

For the stimuli in experiment 2, we do the same comparison, comparing the output of the turn-final condition and the baseline model output. Compared to the results in Table 6 in Table 7 we see fewer significant differences between features in the baseline and turn-final condition. This suggests that our **TURN** model leads to starker differences between the turn-medial condition compared with the baseline along these dimensions.

Table 7: *Significant differences ($p < 0.05$) between BASELINE and TURN turn-final condition for speakers (Wilcoxon signed-ranks test)*

| featnice | 48 | 143 | 156 | 176 | 200 |
|---|---|---|---|---|---|
| Global F0 height | . | . | . | ✓ | . |
| Global F0 slope | . | . | . | . | . |
| Global F0 convexity | . | . | . | . | . |
| Final word F0 height | . | ✓ | ✓ | ✓ | ✓ |
| Final word F0 slope | . | . | . | . | . |
| Final word F0 convexity | . | . | . | . | . |
| Intensity height | . | ✓ | . | . | . |
| Intensity slope | . | . | . | . | . |
| Intensity convexity | . | . | . | . | . |
| Final word log duration | . | ✓ | . | ✓ | ✓ |
| Speech rate (syll/s) | ✓ | . | . | ✓ | ✓ |

## 6. Discussion

In this study, we found that conditioning a model on turn-transition tags leads to increased perception of turn-finality compared to the turn-medial case and the baseline, but that this is speaker-specific. Specifically, speaker-specific differences might just arise from differences in speakers' natural prosodic cues, but also be impacted by the number of each turn-type seen in training. For example speaker 156 learns to sound more turn-final than the turn-medial condition in the **TURN** model, but sounds equally turn-final compared to the baseline. This is potentially due to this speaker having more turn-final IPUs than other speakers in training in the baseline. To test this, future work will compare the turn-medial synthetic speech to the baseline. Future work will also adapt the experiment instructions to ask who is more likely to continue which will help to gain more insight into how instructions might affect results.

Though the amount of data of each class per speaker might affect the comparison between the turn-model and the baseline, we see that for speakers with very few turn-final IPUs in training, they benefited from training on a large amount of turn-final IPUs across other speakers, for example speaker 176 had only 16 turn-final IPUs in training but showed strong learning from other speakers with strong preference for turn-final synthesised audio compared to both the turn-medial and baseline condition. The effect of learning from other speakers can also be seen when we compare the differences in features between the turn-final and turn-medial IPUs in the natural training data, and the differences in the synthetic speech between the turn-medial and turn-final condition with all speakers exhibiting final word lengthening and increased speech rate in the turn-final condition, mirroring results found in corpus studies. This suggests that this was learned corpus-wide as these features were not found in their own productions, though as we will see from the class imbalance, we have few turn-final productions for these speakers.

Initial results suggest that word-final lengthening differences between utterances might be a helpful cue, but possibly only in combination with other factors such as a lowering in f0, again these cues are most likely additive [11] and interact with each other [28]. We also found that the direction of f0 height differences was not always the same across speakers across conditions, this was particularly true for the turn-final synthesis of speaker 143 who had a higher f0 level on the final word compared to the turn-medial synthesis, which may have lead to different turn-finality interpretations. In future work we aim to analyse the turn-finality ratings per experiment per stimulus and correlate these ratings with the individual cues to gain more insight into which specific cues and combinations might be helpful to speakers in judging turn-finality as we found differences in ratings across utterances, even for speakers who showed no significant differences overall between conditions. Future work will also aim to test the ground truth recordings of speakers for their turn taking cues in an experiment and will aim to run experiments with more speakers from the training data.

## 7. Conclusion

In this work, we trained a TTS model with natural conversational data to model turn-taking cues. Overall, we found that our **TURN** model conditioned with the turn-final flag was judged to sound more turn-final than the **TURN** model conditioned with the turn-medial code and more than the baseline, but results are speaker-specific. Interestingly, though our target speak-

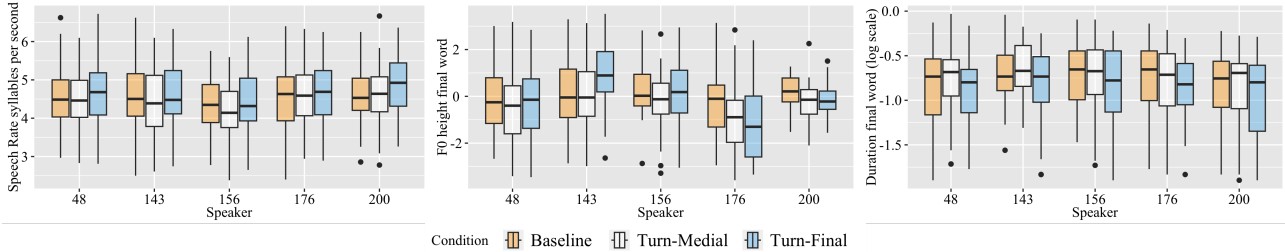

Figure 4: *Speech rate (left), f0 height of final word (centre) and final word duration (right) of TTS output per speaker per condition.*

ers showed large skews in the number of training samples of turn-final utterances to turn-medial utterances, we were able to elicit turn-finality judgements in three of five speakers suggesting turn-finality cues can be learned from large amounts of data of many speakers. We have shown that TTS has potential to be used to analyse which cues listeners and speakers use in the context of turn-taking, though more analysis is needed.

## 8. Acknowledgements

This project has received funding from the European Union's Horizon 2020 research and innovation programme under the Marie Skłodowska-Curie grant agreement No 859588. We thank Niamh Corkey for providing the LJSpeech files.

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
