# OpenReview forum: "Synthesising turn-taking cues using natural conversational data"
_Interspeech.org/2023/Workshop/SSW — SSW12_

### Official Review · Reviewer_Mpkw · 2023-05-22
**First steps towards the generation of turn-taking cues**

**Rating:** 7
**Confidence:** 4

**Review:**

The authors propose to bias a neural TTS (Tacotron2) with a turn-taking label that specifies if the utterance is intended to keep (HOLD) or release (SHIFT) the floor. FastSpeech models are trained using the CANDOR corpus (1656 online conversatiuons between two speakers - > 56hours of exploitable speech data). Turn-Taking bias are identical to speaker embeddings: they are trained and added to each text encoder output before prosodic predictors.

* Key Strength of the paper
First step towards controlable turn-taking generation.

* Main Weakness of the paper
Low MOS ratings. Strong pre-selection process (212 speakers in the training data, 5 tested)

* Novelty/Originality, taking into account the relevance of the work for the SSW audience

* Technical Correctness, is the work technically and/or scientifically solid? Are sufficient details provided to allow any experiments to be reproduced or equivalent experiments run?
Some points are unclear:
1. Why LJSpeech is used?? 23914 CANDOR (41 speakers) vs. 86 LJSpeech utterances seems wuite imbalanced...
2. What are these ambiguous texts? Do they all end whith full stops? What if silence tokens are replaced by adequate punctuations?
3. Why the target speaker corpus training info displayed in Table 2 has such as strong imbalanced distribution (10% turn-final vs 90% turn-medial) compared to numbers given in 3.1.1?

Comments:
1. Why encoding turns as speaker/style bias added to all embeddings? Why not additional tokens similar to punctuation marks, letting the encoder contextualizing the information?

* Quality of References, is it a good mix of older and newer papers? Do the authors show a good grasp of the current state of the literature? Do they also cite other papers apart from their own work?
Refs are adequate

* Clarity of Presentation, the English does not need to be flawless, but the text should be understandable
The text is clear enough

---

> ### Author Response · Authors · 2023-06-28
> **Reply to reviewer 2**
>
> Thank you very much for your review!
>
> Re: Why LJSpeech is used?? 23914 CANDOR (41 speakers) vs. 86 LJSpeech utterances seems wuite imbalanced...
>
> Reply: *We pre-trained on LJ because models trained only on found data tend to have lower quality due to differences in speakers and recording conditions. We did do this in initial experiments and found that the TTS quality was very poor. Following other work (e.g. Szelkely 2019), we decided to pre-train using LJSpeech. We also included a few LJ utterances in post-training (i.e. 86 utterances) which was roughly the same number as most speakers had in the corpus.*
>
> RE: What are these ambiguous texts? Do they all end whith full stops? What if silence tokens are replaced by adequate punctuations?
>
> Reply: *As mentioned in the paper, we remove all punctuation because spoken language does not contain punctuation and automatic punctuation does not always align well with prosodic cues. We could also indeed replace the silence token by a punctuation symbol, but we used the silence tokens returned by the MFA forced aligner. The ambiguous texts were found using the method described in 4.1.1 in the paper.*
>
> RE: Why the target speaker corpus training info displayed in Table 2 has such as strong imbalanced distribution (10% turn-final vs 90% turn-medial) compared to numbers given in 3.1.1?
>
> Reply: *When split into IPUs, natural conversational data often contains a strong imbalance towards non-final IPUs. We therefore constructed the corpus to include as many turn-final IPUs as possible to balance this. This however means that most speakers will also have a strong imbalance (having more turn-medial IPUs). Our data selection aimed to balance these classes across speakers in the whole corpus such that the model would learn patterns across speakers even when a given speaker has fewer turn-final IPUs.*
>
> RE: Why encoding turns as speaker/style bias added to all embeddings? Why not additional tokens similar to punctuation marks, letting the encoder contextualizing the information?
>
> Reply: *We thought about this, but also wanted to potentially use the turn-final or turn-medial flag at other points in the model in future work, so decided to use a label that is passed to the model and not put in the text. This approach has been used in other papers.*
>
> RE: Main Weakness of the paper Low MOS ratings. Strong pre-selection process (212 speakers in the training data, 5 tested).
>
> *Though the MOS ratings are lower than using LJ Speech, the goal of the paper was not to achieve the best MOS but to compare perception across controlled conditions. Our pre-selection process was used because most papers on multi-speaker TTS do not say at all why they chose the speakers they did in evaluation and we wanted to be as transparent as possible by taking speakers who had the most data and sounded the best in the baseline condition (as rated by external listeners).*
>
> Kind regards,
>
> Johannah

---

### Official Review · Reviewer_ktMm · 2023-06-03
**Review of Synthesising turn-taking cues using natural conversational data**

**Rating:** 8
**Confidence:** 5

**Review:**

This study presents compelling research on the synthesis of turn-taking cues, a field that certainly requires more in-depth investigation. The authors' approach of training Text-to-Speech (TTS) voices on ecologically valid data, specifically the CANDOR Corpus of online conversations, is particularly noteworthy. Initially the selected IPUs of length 1-11 sec, removing all questions and filtering out only IPUs of speaking rate 2-6 syll/sec. The training corpus contains 212 speakers where speaker embeddings are used in the training. After selecting the 53 speaker with more than 10 min of data, the authors decided to pre-train on the read speech corpus LJ speech.

The TTS is based on FastPitch with  turn-condition in the input. At the end the five TTS voices with the highest MOS scores were selected for the turn-taking study. In a listening test subject had to judge which of two versions sounded more like the speaker was done speaking. The compared mid-turn with end-of-turn in a first study and a end-of-turn with the baseline trained on all conversational data in the second. The results are that conditioning on turn placement  led  to  increased  perception  of  turn-finality comparing both the turn-medial case and the baseline, but that it is very  speaker-specific. There are some indications that the multi-speaker training have been beneficial for speakers with fewer turn-end IPUs in the training corpus. At the same time there is a large variation across speakers where all speaker but on had higher pitch on the last word in turn-final position.

Some questions to the authors:
The when listening to synthesized texts that are taken from held-out data I get the feeling that the corpus contains a lot of conversations with a stream-of-consciousness-feel that you might get from unplanned telephone conversations. This makes them hard to judge in terms of completeness. It might not always be a strong acoustic cue that makes the other speaker to decide to say something. When you listened to the data, was it easy for you to tell if it is mid-turn or end-of-turn IPU you are listening to?

 One problem with both turn-holding and turn yielding cues is that they overlap, especially in terms of high pitch at the end. You partly solved this be removing a strong turn yielding rise at the end (questions). Turn holding cues can either be planned for, where you use a rising pitch to indicate that there is more to come, and where you mark the end with a falling pitch (turn-yielding cue) – this is what toy have when reading numbers, lists, instructions or retelling events with several parts. Another turn-holding cue is a flat medium pitch and prolongued last syllable (or inserting a filled pause with that prosody). Since you tagged turn-medial and turn-final given who spoke next, it might have been some of the other turn-holding cues listed by Hjalmarsson, Gravano and others at play, such as number of words, grammar or semantics. Or no cue at all - the other speaker just decided to cut in. Did you consider combining the turn-media turn-ending tagging with some kind of prosodic sub-categorization to help the system understand that the prosody is what you want to alter when adding the turn-conditioner to the input text when synthesizing.

Given that the speakers differed so much in prosody, do you think we might benefit from clustering speakers depending on their turn-holding behavior when doing multi-speaker TTS training.

In figure 4 what does final word duration tell us, isn’t that very dependent on which word they happen to end with. You have speech rate and f0 height of final word. Wouldn’t duration of the last syllable say more (related to filled pauses and Gravano-Hirshberg that looked at the last 500 ms). Now you average long and short words where the slowed down speaking rate might only appear on the last few syllables.  Maybe the results become stronger if you limit it to the las syllable(s)

Strengths:
This study, with its expansive scope of building and evaluating multiple voices, as well as performing prosodic analysis on both the TTS corpus and synthesized samples, is indeed impressive.

Limitations:
The results may not be sufficiently robust across different speakers. The test samples often mimic a stream-of-consciousness style, which potentially makes the task of discerning completion of speech demanding for the crowd-workers.

In conclusion, this research provides valuable insights and prompts thoughtful questions, making it a good fit for the SSW conference. I look forward to seeing its presentation and future development.

---

> ### Author Response · Authors · 2023-06-28
> **Replying to reviewer 1**
>
> Thank you very much for taking the time to review our paper.
>
> Re: It might not always be a strong acoustic cue that makes the other speaker to decide to say something. When you listened to the data, was it easy for you to tell if it is mid-turn or end-of-turn IPU you are listening to?
>
> Reply: *We are planning to use our ambiguous test sentences from the evaluation set to test how well listeners can judge turn-finality when they listen to the audio. This will give us a human performance baseline using natural speech. We do not expect listeners to disambiguate perfectly between turn-final and turn-medial because most research on this topic has found prosody to be additive to other linguistic cues.*
>
> Re: Since you tagged turn-medial and turn-final given who spoke next, it might have been some of the other turn-holding cues listed by Hjalmarsson, Gravano and others at play, such as number of words, grammar or semantics. Or no cue at all - the other speaker just decided to cut in. Did you consider combining the turn-media turn-ending tagging with some kind of prosodic sub-categorization to help the system understand that the prosody is what you want to alter when adding the turn-conditioner to the input text when synthesizing.
>
> Reply:  *In this paper, we wanted to take an analysis by synthesis approach to see whether conditioning on these tags indeed led to turn-finality perception or turn-medial perception depending on the tag used, as well as prosodic differences in the resulting synthesis  i.e. we asked: are there prosodic cues that are prevalent across speakers for this feature in the data? In reality it is unlikely that speakers only use prosody to signal a turn-change and that we indeed need other cues e.g. pragmatic and syntactic cues, as well as context. Here our aim was to do a prosodic investigation of turn-taking cues using speech synthesis, while other corpus studies often use statistical testing on various prosodic features.*
>
> Re: Given that the speakers differed so much in prosody, do you think we might benefit from clustering speakers depending on their turn-holding behavior when doing multi-speaker TTS training.
>
> Reply: *Yes, this might be helpful if the goal is to create optimal turn-taking cues or to account for differences in prosody across English accents e.g. some accents always use a final rise in declaratives.*
>
> Re: In figure 4 what does final word duration tell us, isn’t that very dependent on which word they happen to end with. You have speech rate and f0 height of final word. Wouldn’t duration of the last syllable say more (related to filled pauses and Gravano-Hirshberg that looked at the last 500 ms). Now you average long and short words where the slowed down speaking rate might only appear on the last few syllables. Maybe the results become stronger if you limit it to the las syllable(s)
>
> Reply: *Sorry about the confusion. Figure 4 was comparing paired utterances across conditions from the TTS output and not the training data. We therefore were able to use the final word as we had each individual final word spoken in each condition i.e. paired data. It is true though, that given unmatched data we would not use final word as our segment of measurement. We have added clarification in the caption of Figure 4 to say this is analysis of the TTS output.*

---

### Author Response · Authors · 2023-06-28
**Thank you for the reviews**

Dear SSW reviewers,

Thank you for taking the time to review our paper.

We have added a clarification to the caption of figure 4 to clear up confusion.

Kind regards,

Johannah

---

### Decision · Program_Chairs · 2023-06-14

**Decision:**

Accept

**Comment:**

SSW2003 received 45 papers. The acceptance rate is 82%. We are pleased to inform you that your paper has been accepted by the SSW2023 Program Committee. Please read the reviews carefully and submit your camera-ready paper by June 28th. Most reviewers performed a detailed review. Please answer to their questions and consider their comments. Note that camera-ready papers are credited with one extra page to allow authors to consider reviewers’ suggestions. So max 7 pages in total including figures & refs.
The deadline for submitting the revised version (with full non-anonymized authors and refs!) is 28th June.